# Oxidative Corrosion Mechanism of Ti$_2$AlNb-Based Alloys during Alternate High Temperature-Salt Spray Exposure

Wei Chen [1,2], Lei Huang [1,*], Yaoyao Liu [1], Yanfei Zhao [1], Zhe Wang [1] and Zhiwen Xie [2,*]

1   Luoyang Ship Material Research Institute, Luoyang 471023, China
2   School of Mechanical Engineering and Automation, University of Science and Technology Liaoning, Anshan 114051, China
*   Correspondence: 15136307253@163.com (L.H.); xzwustl@126.com (Z.X.)

**Abstract:** This study investigates the corrosion damage mechanisms of Ti$_2$AlNb-based alloys under high temperature, salt spray and coupled high temperature-salt spray conditions. This alloy was analysed in detail from macroscopic to microscopic by means of microscale detection (XRD, SEM and EDS). The results indicated that Ti$_2$AlNb-based alloy surface oxide layer is dense and complete, and the thickness is only 3 μm after oxidation at 650 °C for 400 h. Compared to the original sample, the production of the passivation film resulted in almost no damage to Ti$_2$AlNb-based alloy after 50 cycles of salt spray testing at room temperature. The tests showed that Ti$_2$AlNb alloy shows good erosion resistance at 650 °C and in salt spray. However, this alloy had an oxide layer thickness of up to 30 μm and obvious corrosion pits on the surface after 50 cycles of corrosion under alternating high temperature-salt spray conditions. The Cl$_2$ produced by the mixed salt eutectic reaction acted as a catalytic carrier to accelerate the volatilisation of the chloride inside the oxide layer and the re-oxidation of the substrate. In addition, the growth of unprotected corrosion products (Na$_2$TiO$_3$, NaNbO$_3$ and AlNbO$_4$) altered the internal structure of the oxide layer, destroying the surface densification and causing severe damage to the alloy surface.

**Keywords:** Ti$_2$AlNb alloy; high temperature oxidation; salt spray corrosion; alternating high temperature-salt spray corrosion; corrosion degradation mechanism



## 1. Introduction

Although the traditional nickel-based high-temperature alloys have superior high-temperature oxidation resistance, they are denser and unable to meet the high thrust-to-weight ratio requirements of modern aero-engines [1–4]. Based on the joint action of metal bonds, although conventional titanium-aluminium based alloys (such as α-Ti$_3$Al, γ-TiAl) have outstanding resistance to high temperature oxidation, good creep resistance and organisational stability, the ductility and fracture toughness are poor [5–10]. In order to achieve the service requirements of modern aero-engines and further improve their various properties, the Ti$_2$AlNb alloy was developed by adding Nb to the Ti$_3$Al-based alloy [11,12]. The element Nb not only inhibits the oxidation of Ti, but also increases the activity of Al, enabling the alloy to preferentially produce stable Al$_2$O$_3$ at high temperatures [13–15]. Therefore, Ti$_2$AlNb alloy is expected to be the primary material used for making aero-engine hot-end components. However, frequent starts/stops are commonly accompanied by aeroengines during the service period. In extremely harsh marine environments, alternating high and low temperatures (intermittent operation with room temperature dwell) triggers a more severe electrochemical-thermal corrosion pattern than simple oxidative erosion and salt spray corrosion [16,17]. Therefore, systematic research on the corrosion damage behaviour and degradation mechanism of Ti$_2$AlNb alloy in alternating high and low temperature environments will have considerable theoretical value and broad engineering guidance significance.

In recent years, a series of studies have been carried out at home and abroad on the various properties of Ti$_2$AlNb alloys [15,18–22]. He et al. [18] investigated the oxidation behaviour of a novel-element alloyed Ti$_2$AlNb-based alloy in the temperature range of 650–850 °C. It has been demonstrated that the oxidation rate and mass gain of the Ti$_2$AlNb alloy becomes progressively greater with increasing temperature. The addition of Mo, V, Zr and Si facilitates the selective oxidation of Al to form protective oxides in the alloy, but also the formation of non-protective oxides (such as AlNbO$_4$). Leyens et al. [19] investigated the long-term oxidation behaviour of Ti$_2$AlNb alloys at 650–800 °C. The study concluded that the service time for the alloy to have effective oxygen resistance at 650 °C was 4000 h, which was reduced to 500 h at 700 °C. When the temperature is increased to 800 °C, the alloy shows destabilising oxidation behaviour at 100 h. In addition, Leyens et al. [20] investigated the oxidation behaviour and environmental embrittlement effects of Ti$_2$AlNb alloys in dry and wet air in the range 650–1000 °C. It has been revealed that the oxidation kinetic profile of the alloy below 750 °C is similar to that of conventional titanium alloys. However, the curve shifts gradually to linear in the 800–1000 °C range, where the mixed oxide is mainly TiO$_2$ with Al$_2$O$_3$ and AlNbO$_4$, and internal oxidation is observed at 1000 °C. Dai et al. [21] investigated the high temperature oxidation and hot corrosion behaviour of Ti$_2$AlNb alloys at 923 K and 1023 K. The study indicated that the alloy showed excellent high temperature oxidation resistance and thermal corrosion resistance at 923 K. However, the alloy has poor thermal corrosion resistance at 1023 K and the higher the NaCl content in the mixed salt medium, the more intense thermal corrosion of the alloy.

In summary, researchers from home and abroad have achieved certain research results on high temperature oxidative damage of Ti$_2$AlNb alloys, but there are few reports on the high temperature erosion damage behaviour and performance degradation mechanism triggered by natural salt spray deposition. Although this study has not involved coatings, the alloy is potentially exposed to the same corrosive environment. Therefore, based on the typical coastal conditions of aerospace hot-end components, this paper elaborates on the performance degradation mechanisms of Ti$_2$AlNb alloy under different corrosion conditions using high temperature oxidation tests, salt spray tests and alternating high temperature-oxidation tests. This work provides a comprehensive analysis of the synergy between high temperature oxidative erosion, thermal stress evolution and solid deposited salt corrosion during material damage failure and the damage mechanism, which is also within the technical scope of metal surface engineering. In addition, it has laid an important theoretical foundation for the integrated design, performance control and marine engineering applications of high-performance titanium alloy coatings and structures.

## 2. Materials and Methods

The material chosen for the test was as-rolled Ti$_2$AlNb-based alloy sheet provided by research group of lightweight high-temperature structural materials in tongji university [23,24]. The sheets were cut by wire cutting into blocks of 15 mm, 14 mm and 2.3 mm in length, width and height, respectively. Adequate sample surfaces were ground smooth and flat using 180#, 320# and 600# SiC paper before the tests. These samples were placed in acetone solution and anhydrous ethanol solution to remove surface oils and impurities using ultrasound (see Figure 1).

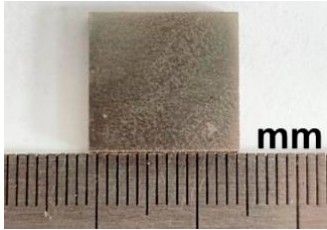

**Figure 1.** Macroscopic surface morphology of the original sample.

High temperature oxidation test was carried out in a high temperature resistance furnace (Shenyang Energy Saving Industrial Electric Furnace Factory, Shenyang, China) as shown in Figure 2a. The samples were subjected to static oxidation at 650 °C for 400 h. The salt spray test was carried out in a salt spray test chamber (Shanghai Meiyu Instruments & Equipment Co., Ltd., Shanghai, China), as shown in Figure 2b. The test is set at 24 h per cycle, with 50 cycles of continuous corrosion. The test chamber solution in the salt spray chamber is a mixed salt solution with a concentration of 5% $\pm$ 0.1% configured from 25 wt.% NaCl + 75 wt.% $Na_2SO_4$. The test chamber temperature was set at 35 $\pm$ 2 °C, the spray preheating temperature was set at 46 °C, the compressed air pressure was set at 83 kPa, the nozzle aperture was 0.5 mm–0.76 mm, the spray volume was based on 0.28 $m^3$ space to atomise 2.8 L of solution in 24 h, and the settling rate was adjusted to 1–3 mL/(80 $cm^2$·h). The pH was adjusted to 3.8–4 with $H_2SO_4$ solution and measured once every 24 h. The high temperature-salt spray test is set at 24 h per cycle. The sample was initially oxidised at 650 °C for 8 h, then cooled to room temperature and placed in a salt spray chamber for 16 h, repeating the above steps to 50 cycles. The specific test design was shown in Table 1.

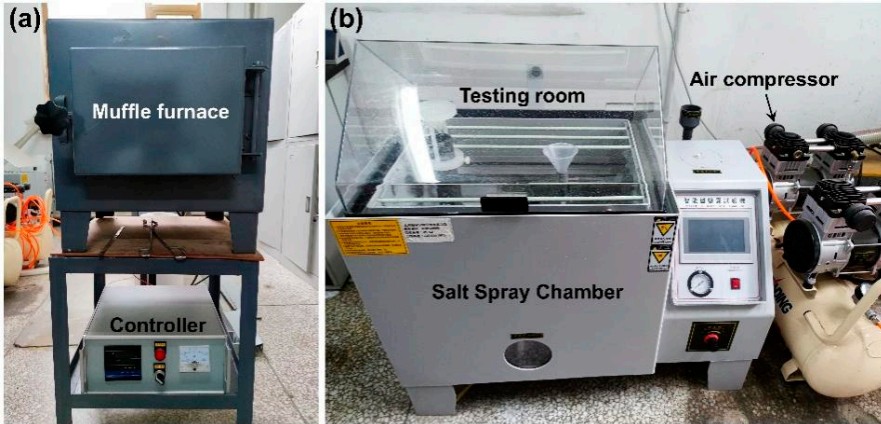

**Figure 2.** Diagram of the test equipment: (**a**) High temperature oxidation test; (**b**) Salt spray test; (**a**,**b**) Alternating high temperature-salt spray test.

**Table 1.** Details of the specific experimental design for Ti$_2$AlNb alloy.

| Sample | Test Type | Time |
|---|---|---|
| D1 | High emperature oxidation test | 400 h |
| D2 | Salt spray test (24 h salt spray corrosion as a cycle) | 50 cycles |
| D3 | Alternating high temperature-salt spray test (8 h of high temperature and 16 h of salt spray corrosion as a cycle) | 50 cycles |

All samples were weighed three times by an electronic balance (Shanghai Sunshine Scientific Instruments Co., Ltd., Shanghai, China) with an accuracy of $10^{-4}$ g. To eliminate the effect of mixed salt on sample mass variation, samples containing salt were washed in boiling deionised water for 10 min before weighing. The test samples were characterised for surface phase composition by X-ray diffraction (XRD, X' Pert Powder, PANalytical B.V., Almelo, The Netherlands) with a Cu-K$\alpha$ source, a scan range of 10°–90° and a scan time of 2 min. Surface and cross-sectional microstructure of test samples were characterised by scanning electron microscopy (SEM, Zeiss $\sum$IGMA HD, Carl Zeiss, Jena, Germany). In addition, energy dispersive spectroscopy (EDS, Zeiss $\sum$IGMA HD, Carl Zeiss, Jena, Germany) was used to further quantify the distribution of elements on the surface and in the cross-section of the sample.

## 3. Results and Discussion

Figure 3 shows the corrosion kinetic curves and surface macroscopic morphology of the test samples. As shown in Figure 3a, sample D1 exhibited a sustained and stable mass variation trend over the course of the test, with an oxidation weight gain of only 0.493 mg/cm² for 400 h. The slope of the curve decreases as the test progresses, showing a parabolic trend. The oxidation rate of the sample during oxidation can be expressed by the following equation:

$$\Delta w^n = k_p t \tag{1}$$

where $\Delta w$ is the oxidation weight gain per unit area (mg/cm²), $n$ is the power index constant, $k_p$ is the oxidation rate constant ($mg^n \cdot cm^{-2n} \cdot h^{-1}$), and $t$ is the oxidation time (h). The results of the linear fit are shown in Figure 3b, and it can be concluded that $k_p$ is about $1.72 \times 10^{-2}$ $mg^2 \cdot cm^{-4} \cdot h^{-1}$, which is similar to the results of Xiang et al. [23].

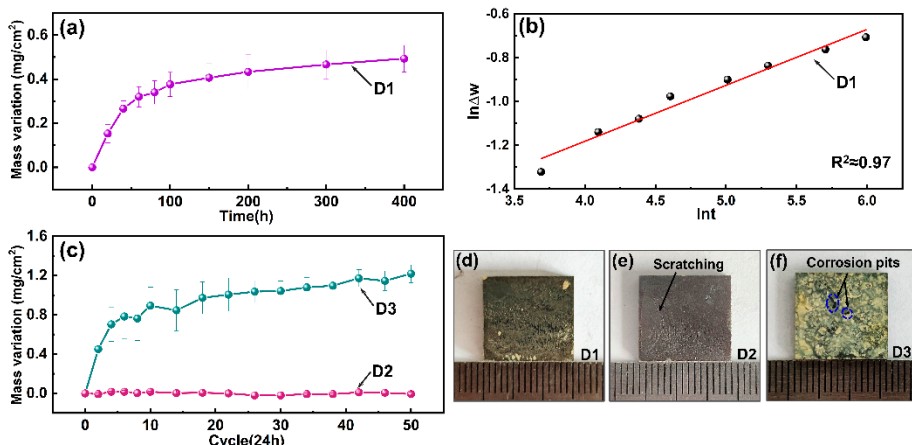

**Figure 3.** Corrosion kinetic curves and surface macromorphology of test samples: (**a,b**) High temperature oxidation test; (**c**) Salt spray test and alternating high temperature-salt spray test; (**d–f**) Surface macromorphology of samples D1, D2 and D3.

In combination with Figure 3d, the overall yellowing of the surface of sample D1 shows obvious brownish-yellow oxidation spots during the 400 h test. However, the surface structure was intact, and no oxide layer cracking or peeling occurred, which indicated that the Ti₂AlNb alloy had excellent oxidation resistance in the 650 °C environment. Compared to sample D1, sample D2 showed extremely small changes in mass throughout the test, and the curve as a whole was close to a smooth straight line. As shown in Figure 3e, although the surface of sample D2 has lost metallic lustre, but the scratches produced when polishing the sample are still visible, indicating that the Ti₂AlNb alloy has excellent corrosion resistance in a salt spray environment. In contrast with samples D1 and D2, the surface of sample D3 suffered a significant corrosion damage in the alternating high temperature-salt spray environment. As shown in Figure 3c, the curve increased rapidly at the beginning of the test, and overall weight increase trend was similar to sample D1, but the final weight increase reached 1.22 mg/cm². In combination with Figure 3f, the surface of sample D3 shows a greenish-yellow colour with tiny corrosion pits, indicating that the surface structure of the Ti₂AlNb alloy was severely damaged in the alternating high temperature-salt spray environment.

Figure 4 shows the XRD patterns of all tested samples. The diffraction peaks on the surface of sample D1 are mainly attributed to the $TiO_2$, $Al_2O_3$ and $Nb_2O_5$, indicating that this alloy underwent a typical oxidation reaction at 650 °C. However, the surface of sample D2 is dominated by the $TiO_2$, indicating that the passivation film produced by the alloy in the salt spray environment has excellent corrosion resistance. In contrast with samples D1 and D2, sample D3 is characterised by the diffraction peaks of obvious non-protective corrosion products ($NaNbO_3$, $Na_2TiO_3$) in addition to the corresponding oxides of Ti, Al and Nb. Combined with Figure 1, it indicated that the alternating high temperature-salt

spray environment accelerated the growth of the alloy oxide layer, and further caused severe corrosion damage to the alloy surface.

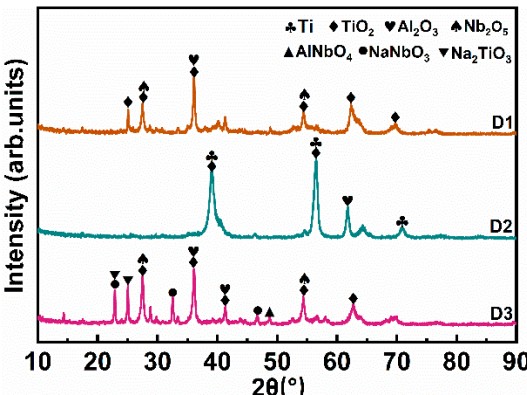

**Figure 4.** XRD patterns of all tested samples.

Figure 5 shows the surface SEM images and EDS spectra of the samples after high temperature oxidation and salt spray tests. As shown in Figure 5a; although an oxide layer had formed on the surface of sample D1 after 400 h testing; but the original scratches caused by SiC paper polishing were still clearly visible; which indicated that the alloy was less oxidised. As shown in Figure 5b; the surface of sample D1 is characterised by a granular morphology; which in combination with EDS analysis is mainly oxides of Ti; Al and Nb (see Figure 5c). In contrast with sample D1; the surface of sample D2 in Figure 5d shows no visible oxide layer or corrosive peeling; and the original abrasion marks caused by the SiC paper are clearly visible. Combined with the EDS analysis in Figure 5f; it indicated that no extensive oxidation occurred on the surface of sample D2; and this alloy showed excellent corrosion resistance during long-term salt spray environment exposure.

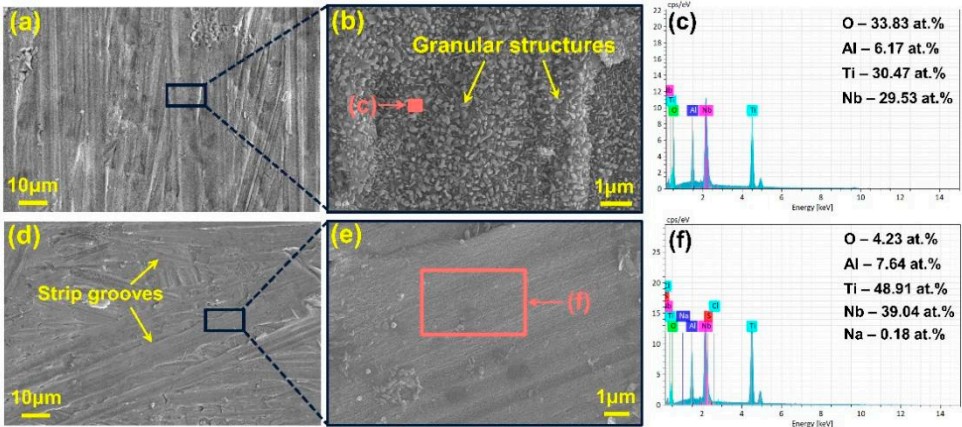

**Figure 5.** Surface SEM images and EDS spectra of the samples: (**a–c**) High temperature oxidation test; (**d–f**) Salt spray test.

Figure 6 shows the SEM images and EDS spectra of the samples after alternating high temperature-salt spray test. As shown in Figure 6a, there was a significant macroscopic shedding of corrosion products on the surface of sample D3. In conjunction with Figure 6b, a great number of slatted fibres grew on the surface of sample D3 after 50 cycles test, indicating that the alternating high temperature-salt spray environment destroyed the surface structure of this alloy, which in turn triggered the corrosion shedding. According to the EDS analysis of Figure 6c,d and the XRD analysis of Figure 4, the surface of this alloy is characterised by the non-protective corrosion products ($NaNbO_3$, $Na_2TiO_3$) in addition to the oxidation products corresponding to Ti, Al and Nb, leading to a change in the denseness

of this alloy surface, and consequently results in longitudinal cracking and flaking of the surface oxide layer.

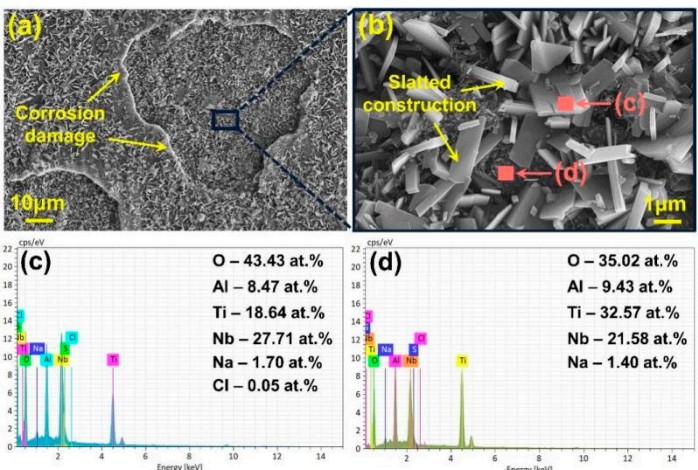

**Figure 6.** Surface SEM images and EDS spectra of the samples after alternating high temperature-salt spray test: (**a,b**) Microstructure of sample D3; (**c,d**) EDS analysis of sample D3.

　　Figure 7 shows the cross-sectional SEM images and EDS spectra of the samples after high temperature oxidation and salt spray tests. As shown in Figure 7a, the oxide layer on the surface of sample D1 is dense and homogeneous, without cracking or flaking, indicating a relatively stable structure. EDS analysis of the line scan shows that the oxide layer contains uniformly distributed Ti, Al and Nb, and the content of Ti is the highest, indicating that the oxide layer is mainly $TiO_2$ accompanied with small amounts of Al and Nb oxides. Compared to sample D1, sample D2 showed no visible structural damage or visible oxide layer on the surface. In conjunction with Figure 7d, there is no significant elemental diffusion and enrichment within sample D2, indicating that the $Ti_2AlNb$ alloy has excellent resistance to salt spray environment exposure.

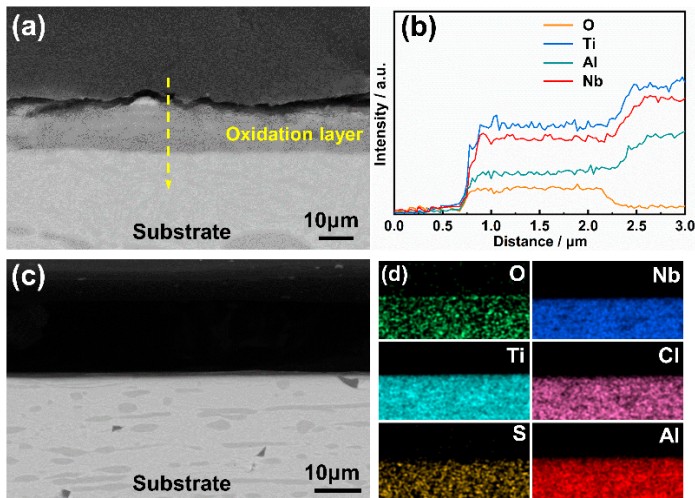

**Figure 7.** Cross-sectional SEM images and EDS spectra of the tested samples: (**a,b**) High temperature oxidation test; (**c,d**) Salt spray test.

　　Figure 8 shows the cross-sectional SEM images and EDS spectra of the samples after alternating high temperature and salt spray test. This alternating high temperature and salt spray environments cause the salt film on the surface of sample D3 to switch back and forth between liquid and solid state. Therefore, the main corrosion that occurs for the alloy in high temperature environments is thermal corrosion caused by settling salt

mixture (75% $Na_2SO_4$ + 25% NaCl). As shown in Figure 8a, the cross-section of sample D3 exhibits a multi-layered structure characteristic, which is probably caused by the internal diffusion of the mixed salt remaining in the salt spray environment. The reaction of the mixed salt with substrate in a high temperature environment produces corrosion products that loosen the surface structure of the alloy. Although the thickness of the corroded layer of sample D3 is relatively homogeneous, but there are some distinct transverse microcracks parallel to the surface oxide layer that exhibit a tendency to damage by delamination and detachment. As shown in Figure 8b, there was significant elemental enrichment and diffusion in sample D3. Combined with the EDS analysis in Figure 8c, the outermost part of the alloy corrosion layer is mainly a mixed oxide layer consisting of $TiO_2$ and a small amount of $Al_2O_3$. However, the area of enrichment of Nb elements is mainly in the middle and lower part of the corrosion layer, which may be related to the growth kinetics of the oxidation products. In addition, the distribution areas of element Na and element Nb show a high degree of consistency on the surface, which combined with Figure 8d, indicating that the non-protective corrosion products ($NaNbO_3$, $Na_2TiO_3$ and $AlNbO_4$) were significant factors in the corrosion damage that occurred on the alloy surface.

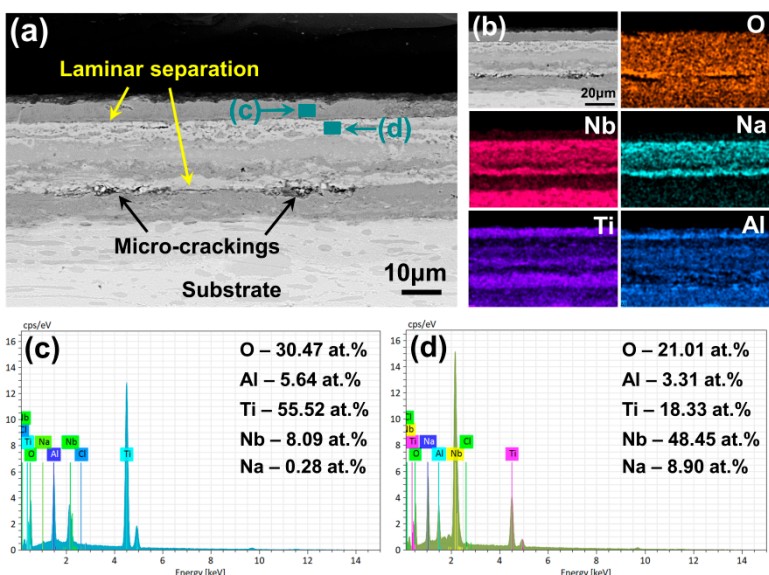

**Figure 8.** Cross-sectional SEM images and EDS spectra of the samples after alternating high temperature and salt spray test: (**a,b**) Microstructure of sample D3; (**c,d**) EDS analysis of sample D3.

The standard Gibbs free energy for the reaction of 1 mol $O_2$ with the elements in the $Ti_2AlNb$ alloy to form the corresponding oxides at 650 °C has been reported: $\Delta G0$ $Al_2O_3$ (−902 KJ/mol) < $\Delta G0$ $TiO_2$ (−759 KJ/mol) < $\Delta G0$ $Nb_2O_5$ (−581 KJ/mol) [25]. However, the growth of oxidation products is not only associated with thermodynamics, but also with growth kinetics. Since the growth kinetics of $TiO_2$ are faster than those of $Al_2O_3$ and $Nb_2O_5$, $TiO_2$ and $Al_2O_3$ are generated almost simultaneously [26].

$$Ti + O_2 = TiO_2 \tag{2}$$

$$4Al + 3O_2 = 2Al_2O_3 \tag{3}$$

As the reaction proceeds, Ti and Al elements on the surface of sample D1 are heavily depleted. Trace amounts of $Nb_2O_5$ are generated on the surface of the alloy due to the high affinity of Nb to oxygen. The prolonged oxidation leads to the reaction of $Al_2O_3$ and $Nb_2O_5$ to form a minor amount of $AlNbO_4$ according to reaction Equations (4) and (5). However, $AlNbO_4$ and $Nb_2O_5$ are not dense enough to prevent the internal diffusion of $O_2$, and mainly the $Al_2O_3$ layer provides protection to the alloy substrate [27]. The test results

show that the weight gain of the $Ti_2AlNb$ alloy at 650 °C oxidation for 400 h is minimal, indicating that the alloy has excellent oxidation resistance in the environment.

$$4Nb + 5O_2 = 2Nb_2O_5 \tag{4}$$

$$Al_2O_3 + Nb_2O_5 = 2AlNbO_4 \tag{5}$$

In contrast to sample D1, sample D2 rapidly produces an extremely dense passivation film on the surface under room temperature salt spray conditions, isolating the external environment from the internal substrate and providing excellent protection for the alloy. As the passivation film is just a few nanometres thick, no specific features are observed by SEM [28–30]. In addition, the majority of metals are inclined to transform from unstable state to a stable state, which means that metal ions and anions in solution can propagate through the membrane, but at a substantially reduced dissolution rate [31,32]. Therefore, based on the protective effect of the passivation film, $Ti_2AlNb$ alloy exhibits excellent corrosion resistance in salt spray environments after 50 corrosion cycles.

However, $Ti_2AlNb$ alloy is severely damaged by alternating high temperature-salt spray corrosion conditions. Figure 9 shows the corrosion damage mechanism for the alternating high temperature-salt spray test. At the initial stage of the test, a composite oxide layer ($TiO_2$, $Al_2O_3$, $Nb_2O_5$) was rapidly formed on the surface of sample D3 in a high temperature environment following reaction Equations (2)–(4) (see Figure 9a) [33]. Upon entering the salt spray environment for the first time from the high temperature environment, salt spray deposits rapidly occur on the surface of the oxide layer of the alloy. The small radius of $Cl^-$ makes it preferentially adsorbed when competing with $O^{2-}$ and $OH^-$ and enters the lattice to occupy the original position of water molecules and oxygen causing accelerated dissolution of the metal [32]. Elemental Al is reported to be extremely poorly resistant to halogen ion corrosion, so that slight localised regional damage begins to occur on the $Al_2O_3$ layer of the alloy surface (see Figure 9b). $Al_2O_3$ can be regarded as $Al_2O_3 \cdot H_2O$ and written as $Al(OH)_3$ in a salt spray environment and the growth of corrosion pits follows the reaction Equations (6)–(9) [34]:

$$Al(OH)_3 + Cl^- \rightarrow Al(OH)_2Cl + OH^- \tag{6}$$

$$Al(OH)_2Cl + Cl^- \rightarrow Al(OH)Cl_2 + OH^- \tag{7}$$

$$Al(OH)Cl_2 + Cl^- \rightarrow AlCl_3 + OH^- \tag{8}$$

$$AlCl_3 \rightarrow Al^{3+} + 3Cl^- \tag{9}$$

when sample D3 is re-introduced to the high temperature environment, original liquid salt film transforms into solid salt film adhering to the surface of the substrate. As shown in Figure 9c, the predominant corrosion occurring on sample D3 during the period is thermal. The mixed salts (NaCl and $Na_2SO_4$) are in the molten state at 650 °C and can diffuse internally through defects and loose parts of the oxide layer where eutectic reactions will occur according to reaction Equations (10)–(14) to produce $Cl_2$ [27].

$$Na_2SO_4 = Na_2O + SO_3 \tag{10}$$

$$2SO_3 = 2SO_2 + O_2 \tag{11}$$

$$2SO_3 = S_2 + 3O_2 \tag{12}$$

$$2NaCl + O_2 + SO_2 = Na_2SO_4 + Cl_2 \tag{13}$$

$$4NaCl + 4O_2 + S_2 = 2Na_2SO_4 + 2Cl_2 \tag{14}$$

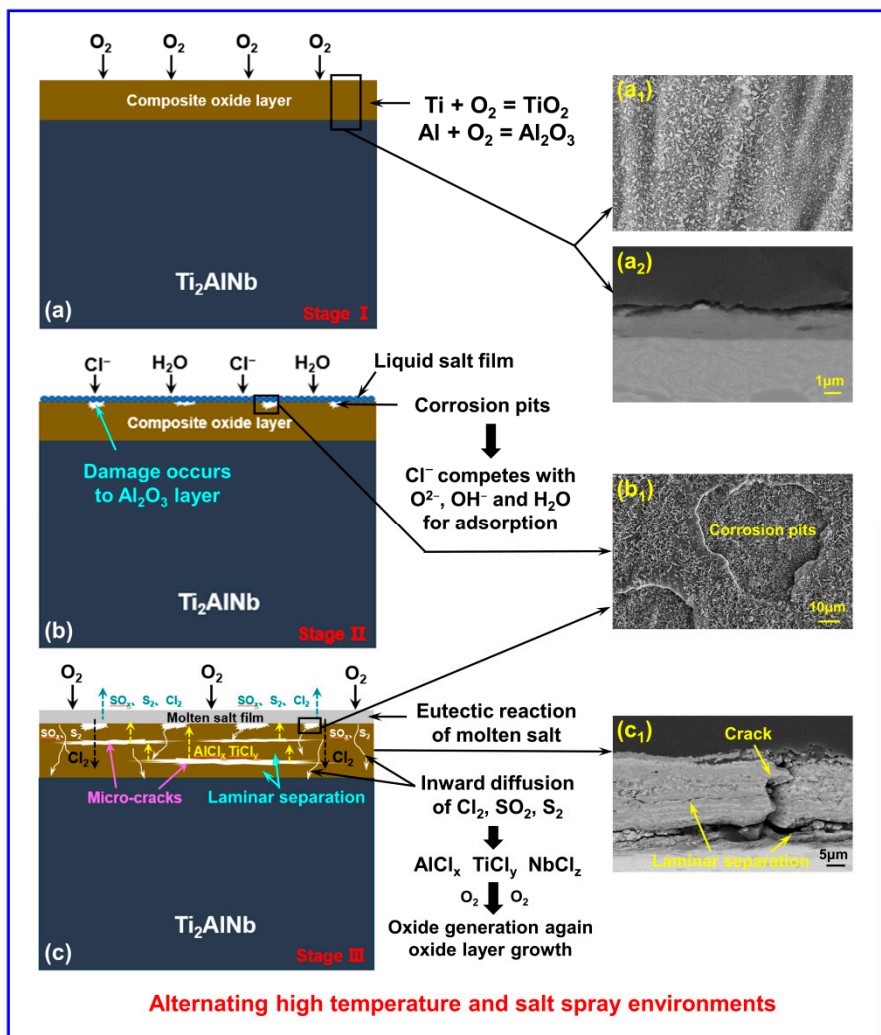

**Figure 9.** Corrosion degradation mechanisms in alternating high temperature-salt spray test: (**a–c**) Mechanistic diagram of the test procedure; (**a1**,**a2**) Microstructure of the alloy after oxidation test; (**b1**,**c1**) Microstructure of the alloy after alternating high temperature-salt spray tests.

The corrosion caused by the salt spray destroyed the compactness of the oxide layer on the surface of the sample, resulting molten salts and reaction products penetrating inwards into the oxide layer. The $Cl_2$ produced by the reaction with Ti, Al and Nb inside the oxide layer produce the corresponding chlorides according to reaction Equations (15)–(17) [35,36].

$$Ti + 2Cl_2 = TiCl_4 \tag{15}$$

$$2Al + 3Cl_2 = 2AlCl_3 \tag{16}$$

$$2Nb + 5Cl_2 = 2NbCl_5 \tag{17}$$

The chloride has a low boiling point and evaporates rapidly at 650 °C, causing a large number of micropores to be created within the oxide layer and longitudinal micro-cracks to be derived from the aggregation effect (see Figure 8a). In addition, the creation of holes and cracks provides a channel for the internal diffusion of external $O_2$. The reaction of $O_2$ with the chloride causes re-oxidation of the surface, which in turn accelerates the growth of the oxide layer, the primary reason why the thickness of the oxide layer of sample D3 is thicker than that of sample D1. For specific reaction Equations see (18)–(21) [37].

$$4TiCl_3 + O_2 = TiO_2 + 3TiCl_4 \tag{18}$$

$$TiCl_4 + O_2 = TiO_2 + 2Cl_2 \tag{19}$$

$$4AlCl_3 + 3O_2 = 2Al_2O_3 + 6Cl_2 \tag{20}$$

$$4NbCl_5 + 5O_2 = 2Nb_2O_5 + 10Cl_2 \tag{21}$$

Therefore, the above reaction constitutes a self-cyclic reaction, where $Cl_2$ acts as a catalytic carrier to accelerate the reaction, resulting in a gradual thickening of the oxide layer, but decreasing in densities. The molten salt and reaction products ($Na_2O$, etc.) continuously penetrate through the loose oxide structure and generate a large number of unprotected corrosion products with $TiO_2$, $Al_2O_3$ and $Nb_2O_5$ following reaction Equations (22)–(25) [24,38]. In addition, the generation of $AlNbO_4$ from $Nb_2O_5$ and $Al_2O_3$ further reduces the denseness of the oxide layer. As a result, the high thermodynamic growth stress difference between the loose oxide structure and the corrosion products causes the oxide layer to continue to flake off with increasing test time [21].

$$Nb_2O_5 + Na_2O = 2NaNbO_3 \tag{22}$$

$$TiO_2 + Na_2O = Na_2TiO_3 \tag{23}$$

$$Al_2O_3 + Nb_2O_5 = 2AlNbO_4 \tag{24}$$

$$4NaCl + 2Al_2O_3 + O_2 = 4NaAlO_2 + 2Cl_2 \tag{25}$$

when the sample is re-entered into the salt spray environment, the solid salt film dissolves and the surface is once again covered by a liquid salt film, causing electrochemical corrosion to occur making the surface corrosion pits progressively more severe. With the cycling of test and the switching between solid/liquid salt films, this resulted in severe corrosion degradation of the sample surface, complete failure of the protective effect of the oxide layer and severe damage to the alloy.

## 4. Conclusions

This study investigated the corrosion damage behaviours of $Ti_2ANb$ alloys during high temperature, salt spray and alternating high temperature-salt spray environment exposures. $Ti_2AlNb$ alloy exhibited excellent oxidation resistance at 650 °C. The weight increased only 0.493 $mg/cm^2$ after 400 h oxidation test. The oxide layer was dense and uniform, and the oxidation products were mainly $TiO_2$ and $Al_2O_3$. Due to the generation of a dense passivation film, $Ti_2AlNb$ alloy showed an excellent resistance to salt spray corrosion after 50 cycles of salt spray corrosion with a near straight line quality change curve and almost no surface damage. However, after 50 cycles of corrosion in an alternating high temperature-salt spray environment, $Ti_2AlNb$ alloy increased in weight by up to 1.22 $mg/cm^2$, almost three times the weight gain of the high temperature oxidation test sample. The adhesion of the solid salt film caused the $Cl_2$ generated by the eutectic reaction to penetrate into the interior of the oxide layer, resulting in the volatilisation of gaseous chloride and the derivation of a large number of micro-pores. The internal diffusion of molten salt further induced the growth of unprotected corrosion products ($Na_2TiO_3$, $NaNbO_3$ and $AlNbO_4$) and destroyed the densification of the alloy surface. Frequent changes in corrosion media caused by alternating high and low temperatures seriously damaged the surface integrity of this alloy and greatly shorten its service life.

**Author Contributions:** Conceptualisation, L.H. and Z.X.; methodology, Y.Z.; investigation, W.C. and Y.L.; supervision, Z.W. All authors have read and agreed to the published version of the manuscript.

**Funding:** This study was supported by the University of Science and Technology Liaoning Talent Project Grants (601011507-07).

**Institutional Review Board Statement:** Not applicable.

**Informed Consent Statement:** Not applicable.

**Data Availability Statement:** No new data were created or analysed in this study. Data sharing is not applicable to this article.

**Conflicts of Interest:** The authors declare no conflict of interest.

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
