# Peer review of "Oxidative Corrosion Mechanism of Ti2AlNb-Based Alloys during Alternate High Temperature-Salt Spray Exposure"

_coatings, doi:10.3390/coatings12101374_

Round 1

Reviewer 1 Report

The authors provide a valuable investigation into the oxidation and corrosion of the useful alloy Ti2AlNb.  However, their report is not as clear as it should be on a number of issues which the authors are requested to address. 

The description of the samples used in their experiment is impressively devoid of essential details.  The description of the samples as 'lumpy' raises a concern.  How could this have occurred?  Where did the samples come from: for instance were they produced by a powder process, or vacuum arc refining, or electron beam or plasma hearth refining? Perhaps they were small button samples from a laboratory arc melter, involving turning the samples over repeatedly to melt each side. 

The length of the samples means nothing without the width and height measurements, or length and diameter etc. 

The very old, dated nomenclature 'sand paper' and 'sanding down' should be avoided. Sand is too soft to make any impression on most alloys, and the dust is hazardous.  The modern equivalents are now 'SiC paper' and the concept of abrasion, so that parts can be abraded with SiC paper.  

Similarly out-dated concepts are the unhelpful mesh numbers to describe the sizes of powders.  It is normal to avoid these old systems and use the simple and accurate system of average sizes of particles measured in micrometers. A quick check under a microscope will give a sufficiently good value to quote.

Figure 3 is curiously spit into two separate graphs, which is not helpful to readers to compare the three curves.  All three curves should be drawn on one graph - which is extremely easy to do, and highly informative to a reader.

Throughout the paper the authors repeatedly mention cracking, while never making it clear whether (i) the cracks are in the oxide or in the metal matrix; and (ii) they do not mention whether the cracks are parallel to the surface of the interface, or at right angles. 

Line 198. What does 'settling salt mixture' mean?  

Line 200. Once again, the 'internal diffusion' in oxide or metal matrix?  (some of these questions are answered elsewhere in the text, but they are not easy to find. It is therefore much more helpful to readers to give the information immediately.  As authors, you know this, but overlook to say it in your text.

Line 204. Significant elemental enrichment and diffusion in D3'. Once again in oxide or metal matrix?

Line 253 and following: 'Pitting damage' is the phrase used by the authors, but this reviewer is of the opinion that this is a misleading use of this term.  A corrosion pit as this reviewer understand the concept, is a highly localized but deep pit, often roughly hemispherical in shape.  The so-called pits in this report are nothing like this. I think they should not be referred to as pits, but referred to a pieces of spalled-off oxide.  They are uniformly shallow and reasonably extensive in area (nothing like a corrosion pit) and seem confined to the oxide layer (all corrosion pits result from attacks on the metal matrix.  They are nearly always linked to what appears to be stress corrosion cracks which appear to be nucleated from the base of the pit, and extend deeply into the metal matrix). A change of terminology is definitely required. 

Reviewer 2 Report

Review of paper titled “Oxidation corrosion mechanism of Ti2AlNb alloys during alternative high temperature – salt spray exposure” for MDPI Coatings

In this article, the authors studied the corrosion kinetics of a Ti2AlNb alloy. High temperature and salt spray corrosion tests were employed and combined. The paper is interesting and novel. It is publishable subject to revision.

1.The authors must clarify why they decided to submit their paper to Coatings. There is no coating on the alloy. Only a bulk material was studied. If the passive layer formed during oxidation is meant to play the role of "a coating" that will eventually protect the material against further oxidation, the authors should explain it in the introduction.

2.The manufacturer and chemical composition of the Ti2AlNb alloy should be defined in the Materials and Methods section. Any possible trace elements should be specified.

3.The authors speak about “marine service conditions of aerospace hot-end components” (lines 72-73). However, a more appropriate term would be coastal conditions. Marine service conditions are used for ships.

4.The authors should explain why sodium sulfate was added to the salt spray tests. Have you tried to model a heavily polluted coastal atmosphere?

5.The cycles should be clearly defined in Table 1 as they differ between the experiments. The salt spray tests included 24 hour cycling at room temperature, while the alternating corrosion tests had an 8 hour high temperature exposure followed by 16 hour salt spray in each cycle. The definition of cycles should be given in Table 1.

6.You should get a parabolic rate constant from the data in Fig. 3a and compare it with literature.

7.There are several unidentified peaks at 40 – 50° (2-theta) in Fig. 4 (XRD patterns). The authors should assign them.

8.Figures 5 and 6 have different magnifications. As such, it is difficult to compare the morphology and grain size of the oxides grown under different conditions. The authors should provide images with identical magnification.

9.The discussion should use more rigorous terms:

Line 220: the growth kinetics of TiO2 was FASTER instead of “higher”

Line 226: Trace amounts of Nb2O3 were generated due to the high AFFINITY of Nb to oxygen instead of the “high thermodynamics of Nb growth”.

Line 266: The pitting destroyed the COMPACTNESS of the oxide scale instead of the “denseness”

10.Please have your paper proof-read by a native speaker familiar with materials science terminology before resubmitting.

Round 2

Reviewer 2 Report

Authors answered my comments. The paper can be accepted for publication.